# Scalability of Cyber-Physical Systems with Real and Virtual Robots in ROS 2

**DOI:** 10.3390/s23136073

**Published:** 2023-07-01

**Authors:** Francisco José Mañas-Álvarez, María Guinaldo, Raquel Dormido, Sebastian Dormido-Canto

**Affiliations:** Department of Computer Sciences and Automatic Control, Universidad Nacional de Educación a Distancia (UNED), Juan del Rosal 16, 28040 Madrid, Spain; fjmanas@dia.uned.es (F.J.M.-Á.); mguinaldo@dia.uned.es (M.G.); raquel@dia.uned.es (R.D.)

**Keywords:** multi-robot system, digital twin, ROS 2, formation control

## Abstract

Nowadays, cyber-physical systems (CPSs) are composed of more and more agents and the demand for designers to develop ever larger multi-agent systems is a fact. When the number of agents increases, several challenges related to control or communication problems arise due to the lack of scalability of existing solutions. It is important to develop tools that allow control strategies evaluation of large-scale systems. In this paper, it is considered that a CPS is a heterogeneous robot multi-agent system that cooperatively performs a formation task through a wireless network. The goal of this research is to evaluate the system’s performance when the number of agents increases. To this end, two different frameworks developed with the open-source tools Gazebo and Webots are used. These frameworks enable combining both real and virtual agents in a realistic scenario allowing scalability experiences. They also reduce the costs required when a significant number of robots operate in a real environment, as experiences can be conducted with a few real robots and a higher number of virtual robots by mimicking the real ones. Currently, the frameworks include several types of robots, such as the aerial robot Crazyflie 2.1 and differential mobile robots Khepera IV used in this work. To illustrate the usage and performance of the frameworks, an event-based control strategy for rigid formations varying the number of agents is analyzed. The agents should achieve a formation defined by a set of desired Euclidean distances to their neighbors. To compare the scalability of the system in the two different tools, the following metrics have been used: formation error, CPU usage percentage, and the ratio between the real time and the simulation time. The results show the feasibility of using Robot Operating System (ROS) 2 in distributed architectures for multi-agent systems in experiences with real and virtual robots regardless of the number of agents and their nature. However, the two tools under study present different behaviors when the number of virtual agents grows in some of the parameters, and such discrepancies are analyzed.

## 1. Introduction

Cyber-physical systems (CPSs) can be considered a new generation of digital systems and their impact on the acceleration of technological progress is huge. A CPS arises from a tight integration of a cyber system and a physical process. It is defined as a system that deeply joins the capacity of computing and communication to control and interact with a process in the physical world [1]. Through the feedback loop between computing and the process, real-time interactions are increased with the physical system to monitor or control the physical entity in a secure, efficient, and reliable way in real time.

Examples of CPSs include applications in many areas such as aerospace, transportation, manufacturing, automotive, etc. [2]. The heterogeneous nature of CPSs requires advanced knowledge in several disciplines for their design and construction. For instance, the correct operation of applications with huge potential impacts on the daily lives of many people such as the so-called Human-in-the-Loop Cyber-Physical Systems (HiLCPSs). It needs not only technical knowledge to design the interface between the human and the autonomous components but also control competences to design strategies for these human-in-the-loop systems [3,4]. Currently, most CPS researchers mainly focus on system architecture, information processing, and software design [5,6].

The building of CPSs in mobile robotics is particularly interesting [7]. In designing and implementing advanced robotic heterogeneous hardware platforms for controlling an ever-larger multi-agent system, several challenges appear [8]. Integration of all hardware devices and the requirement of reliable, concurrent hardware access with real-time constraints makes the design and implementation of these systems a difficult task. In particular, when the number of agents increases, several problems related to control or communication arise due to the lack of scalability of existing solutions. Developing and comparing tools that allow control strategies evaluation of large-scale systems is necessary. The main contribution of this paper is oriented towards this purpose.

Usually, before working the system in a real-world scenario, tests under controlled environments to reduce possible unexpected behaviors must be accomplished. In this regard, different alternatives are available: implementing through virtual environments using robotic simulation tools; using a physical environment that allows validating developments in a real platform; or making use of hybrid platforms that combine real and virtual agents [9]. Whatever scheme is chosen, a robotic simulation tool is always used in the first steps before real implementation, making the right selection of the most suitable tool an important task for the scientific and academic community.

In this paper, it is considered that a CPS is formed by a heterogeneous multi-robot system (MRS) that cooperatively performs a formation task through a wireless network. In recent years, the number of simulation tools and technologies available for the development of MRS platforms has grown substantially. Digital twins (DTs) and digital shadows [10,11,12,13] are among those technologies that perfectly integrate the principles of CPSs and their capability to manage information from a virtual world and to allow data collection and exchange in real time with the physical system. Augmented reality (AR) has also been used as a support tool to integrate physical and digital environments [14]. Unlike virtual reality (VR), which generates representations of the physical world, AR, mainly focusing on collaborative robotics, allows us to superimpose digital information on elements of the physical world. Mixed reality (MR) involves a combination of VR and AR [15,16,17]. An MR based system interacts and manipulates both physical and digital elements and environments by a completely bidirectional communication [18,19,20]. In this way, MR has great potential to solve classic mobile robotics problems in an intuitive and efficient way [21]. MR technology has been used in different applications in robotics [22,23]. For instance, it has been explored in social robotics to enhance a robot with limited expressivity [22].

Robotic Park [9,24], a recently developed complete heterogeneous flexible, and easy-to-use indoor platform to perform MRS experiments in the Department of Computer Sciences and Automatic Control of UNED, is the experimental framework where our tests are carried out. It supports virtual, real, or hybrid scheme experiences through the two hybrid frameworks available that enable combining physical (and their DT) and virtual agents in MR experiences. As the DTs are indistinguishable from the real agents, their nature is indistinguishable when integrated into ROS 2.

Among the most popular MRS simulators are Gazebo [25], V-REP (Coppeliasim now) [26], Webots [27], or MVSIM [28]. As they were not built as general-purpose tools it is difficult to compare them. In the literature, several works are found comparing different simulators pointing out the strengths and weaknesses of each one [29,30,31]. Comparisons on features such as programming languages, supported OS, accuracy, open-sourced or related to robots, sensors, or actuators supported are available [32,33]. Gazebo, a versatile open-source simulator with a large community of developers, is the reference software tool for robotics research widely used over time. V-REP is simpler to manage and makes it simpler to edit objects, but a license is required [34]. Recently, Audonnet et al. studied a robotic arm manipulation under different simulation software compatible with ROS 2 [35]. In this work, they give a comparison between Ignition and Webots in terms of stability and analyze PyBullet and Coppeliasim attending the resources usage. No best simulation software is determined overall. In [36] a comparison of four simulation environments for robotics and reinforcement learning can be found. Pitonakova et al. [37] compared the functionalities and simulation speed of Gazebo, ARGoS, and V-REP. If we focus on quantitative comparisons, few works have been reported for the most used robot simulation tools. In [31] a quantitative comparison among CoppeliaSim, Gazebo, MORSE, and Webots on the accuracy of motion for mobile robots can be found. The main motivation of this work is to evaluate the system’s performance when the number of agents increases. The objective is to carry out a comparative analysis of two popular robotic simulators, Gazebo and Webots, determining the simulator that allows working with a larger number of agents, what the resource consumption is for each one, and to determine the limit in the possible combination of real and virtual agents up to which the Real Time Factor drops. Moreover, an analysis of convergence times for both simulators to achieve the desired formation using an event-based control strategy has been performed. Both a quantitative and objective comparison of the two simulators is shown.

For that purpose, two different frameworks within Robotic Park have been developed with the open-source tools Gazebo and Webots. Both tools enable MRS experiences combining both physical and virtual agents in a realistic scenario regardless of the number of agents and their nature, allowing scalability tests with no extra cost when a significant number of robots operate in a real environment. In this way, several experiences that address an event-based control strategy for rigid formations varying the number of agents (either aerial robot Crazyflie 2.1 or differential mobile robots Khepera IV) are carried out. The required formation is defined by a set of desired Euclidean distances among neighbors. Several metrics are used to analyze the scalability of the tools: the formation error, CPU utilization as a percentage, and the ratio between the real time and the simulation time. The results illustrate the scalability of Robotic Park as experiences can be conducted with a few physical robots and a large number of virtual robots that mimic the real ones. Moreover, the use of ROS 2 facilitates the addition of new agents, a decentralized communication system, and greater integration with simulation environments.

The paper is organized as follows. Section 2 describes in detail the experimental environment and its components. Section 3 presents the main features of the MRS focused on formation with distance constraints used and defines the experiments carried out to perform the comparison between Gazebo and Webots. Section 4 discusses experimental results using the platform. Finally, Section 5 ends the paper with conclusions and suggestions for future works.

## 2. Materials and Methods

### 2.1. Experimental Platform

The experiences conducted in this work have been performed over Robotic Park [9] (Figure 1a). This platform is focused on indoor multi-robot experiences with heterogeneous agents. The workspace has a volume of 3.6 m × 3.6 m × 2 m. This space allows the operation of a relatively large number of small robots in formation and navigation tasks. Different positioning systems are available using different technologies (Motion Capture, UWB, Infrared, etc). This enables combining different agents, as some of them can only use a single system. Among the different robots available on the platform, in this work, the micro aerial robots Crazyflie 2.1 (Figure 1b) and mobile robots Khepera IV (Figure 1c) are used. The Crazyflie 2.X is an open-source platform developed by Bitcraze [38]. It is suitable for indoor experimentation due to its small size (92 mm × 92 mm × 29 mm), low mass (27 g), and inertia. Khepera IV is a wheeled mobile robot designed by K-Team [39,40]. It is specially designed to work on hard and flat indoor surfaces.

### 2.2. Simulators

Gazebo and its integration with ROS/ROS 2 distributions is one of the most widely used frameworks in robotics [25]. It is a multi-platform open-source software with a large number of libraries developed by users to support a wide range of sensors, actuators, robots, etc. It allows the development of multi-robot experiences with easy management of the loading and unloading of assets in the environment. Gazebo allows the implementation of distributed simulations and adjustable real-time factor performance. It supports different physics engines: Bullet, Dynamic Animation and Robotics Toolkit (DART), Open Dynamics Engine (ODE), and Simbody. In addition, Gazebo uses the Universal Robot Description Format (URDF) and the Simulation Description Format (SDF), the two most widely used representation formats for robotics simulations. Figure 2a shows the twin of the Robotic Park platform developed with Gazebo.

Webots is a simulation platform with a long history since its launch in 1996 by Dr. Oliver Michel at the Swiss Federal Institute of Technology EPFL in Lausanne [27]. Since 1998 Cyberbotics Ltd. is the main maintainer of the tool. It is a multi-platform open-source software with wide use in industry and research. As a physics engine, it exclusively uses ODE and it supports C++, Python, Java, and Matlab as programming languages. It has recently added a ROS 2 bridge among its features. It has a more user-friendly interface than other tools, allowing users to manually interact in real time with the robots. This facilitates the use of the tool for non-experienced users, such as students. Finally, Webots supports the PROTO format for real objects/robots representation. It is possible to use a short URDF format file to fix robots’ parameters (they could be parameters defined by users or ROS 2 plugins’ characteristics, such as sensor update rate). Figure 2b shows the developed twin of the Robotic Park platform in Webots.

### 2.3. ROS 2

ROS is a set of open-source software libraries and standard tools that help in building robot applications [41]. Its most recent version, ROS 2, includes new features that make ROS easier to learn and use. This fact encourages new users who until now were reticent to ROS. These improvements enable a more efficient communication protocol with a better real-time performance than ROS that allows distributed architectures, an adaptation to the most recent language libraries, such as Python 3, and native multi-platform development. This last feature brings its use closer to Mac or Windows users, two systems with wide presence among non-professional users.

In Robotic Park, ROS 2 is used as a link between all agents in the system (real and virtual). It allows easier integration of new agents without updates to already installed agents. Each element (robots, positioning systems, cameras, etc.) has a node that serves a single, modular purpose in the system. On the one hand, a ROS 2 node is created for the positioning systems that are responsible for the publication of the global position of the robots in their corresponding topics (geometry_msgs/Pose type). On the other hand, the nodes associated with the robots are in charge of subscribing to relevant topics for them, such as their global position from the external positioning systems. These same nodes publish the robots’ internally estimated position from their global position and other internal localization systems, such as IMUs or odometry. The nodes of those agents that are connected in the formation subscribe to these topics to close the coordination control loop. A DT model recreates real robots in the virtual environment with identical characteristics to the real system. DTs run on the simulation software and have a twofold purpose. First, they allow complementing the sensors of the real robot, such as distance sensors. This enables our experiences to emulate the presence of virtual robots in the physical environment and to make obstacle avoidance algorithms respond as if all robots were real, and secondly, they can also be used for fault detection. For instance, in case of the loss of a real robot, it could be temporarily replaced by a digital twin, to avoid triggering failures in the movement of the rest of the agents.

In the designed ROS 2 architecture, each robot is defined within a namespace where all the nodes necessary for its operation are grouped. For instance, Figure 3 shows a subset of two robots’ namespaces of these experiments. The robot “*i*” is a physical robot with its DT and the robot “i+1” is just a virtual robot. A driver node is defined for each robot, which is responsible for the communication with sensors and actuators. DTs in a ROS network are indistinguishable from real robots since they can use the same nomenclature and types of nodes and topics.

### 2.4. Computational Resources

The setup of this work is developed on a laptop with a centralized architecture. In this way, all the nodes and the simulation software run on the same CPU. It is an HP ZBOOK POWER G7 Mobile Workstation 15.6″ with an Intel core i9-10885H, DDR4 3200 64 GB RAM, 2 TB SSD memory, and NVIDIA TU117GLM [T1200 Laptop GPU] and Intel Corp. TigerLake-H GT1 as the graphics card. Ubuntu 22.04 is the operating system installed on the PC with kernel version 5.19.0-41-generic. To read the CPU consumption, the Python library *psutil* in a ROS 2 node is used.

## 3. Problem Formulation and Experiments

In this section, we first present the control objective and implemented architecture to perform the experiments, and secondly, the specifications for the experiments and the analyzed metrics to study the scalability are described.

### 3.1. Control Architecture

The main goal of this work is to explore the limit in the number of agents supported by two of the main simulators used in the field of control and robotics. For that purpose, we consider a formation control problem in which the complexity of the formation basically depends on the number of agents, *N*. Specifically, the desired formation is defined in such a way that the agents should be uniformly distributed over a semi-spherical virtual surface. The ground level of the dome (z=0) is composed of mobile robots, and the rest of the robots are drones. Figure 4 shows an example of the desired 3D formation and the projection over the XY-plane for N=50 and a semi-sphere centered at (0,0,0) and a radius R=2 m. The drones are distributed in rings of different height, each of which is drawn with a different color. The uniform distribution of points over a sphere is a classical mathematical problem that is difficult to solve analytically, and recent approximate solutions have been proposed [42]. The MRS is modeled in terms of a graph G(V,E), where $V=\left\{v_{1},\cdots ,v_{N}\right\}$ is a finite set of *N* vertices representing the nodes or agents and E⊆V×V is a finite set of edges, representing the communication links between them. The graph G is defined in such a way that the formation is ensured to be rigid [43] and, as a result, target values are generated for the inter-distances between any two nodes $v_{i}$ and $v_{j}$ that are connected in the graph ($v_{i}v_{j}\in E$).

In Robotic Park a hierarchical control structure with three levels is available for experiences. It includes a position controller at the lower level, a coordination controller, and path planning at the upper level (see Figure 5). The solution to the formation control problem with collision avoidance proposed in this paper does not involve the upper level. Thus, the intermediate and the lower level participate as described below.

At the lower level, the position controllers follow a distributed architecture and are implemented onboard each robot. They close the control loop with signals generated by internal estimators from their sensors’ data and the external positioning systems (they have to ensure the stability of the closed-loop response) and work at the highest frequency (between 100 and 500 Hz). Different control architectures have been implemented on the two types of robots used in this work [9]. For instance, the Kheperas IV use two controllers focused on position and orientation, and the Crazyflie is controlled using a two-level cascaded PID controllers scheme. Moreover, at this level, an additional term is included to avoid collisions between robots. This control term is derived from repulsive potential fields as follows: (1)$$U_{k}=\left\{\begin{array}{cc} \frac{1}{2}\eta {\left(\frac{1}{d_{k}}-\frac{1}{d_{0}}\right)}^{2} & if\;\;d_{k}\leq d_{0}\\ 0 & if\;\;d_{k}>d_{0} \end{array}\right.$$
where $U_{k}$ denotes the repulsive potential of sensor *k*, $d_{0}$ is a threshold that activates the repulsive potential, $d_{k}$ is the value of the distance between the sensor and the obstacle, and η is a constant that characterizes the field. Then, the resulting repulsive force $F_{k}$ is defined by: (2)$$F_{k}=-\nabla U_{k}=\left\{\begin{array}{cc} \eta \left(\frac{1}{d_{k}}-\frac{1}{d_{0}}\right)\frac{1}{d_{k}^{2}}\frac{p_{k}-p_{o}}{d_{k}} & if\;\;d_{k}\leq d_{0}\\ 0 & if\;\;d_{k}>d_{0} \end{array}\right.$$
where $p_{k}-p_{o}$ is the relative position between the robot and the obstacle. Hence, the sum of all repulsive forces is $F=\sum_{k}F_{k}$, which has an impact on the goal position according to the following expression: (3)$$u^{oa}=h\cdot v\cdot \frac{F}{∥F∥}$$
where $u^{oa}$ is the deviation of the goal position signal received from the coordination level, *h* is the period of the controller, and *v* is a constant velocity.

The intermediate level is responsible for coordinating the movements between agents to ensure that the desired formation is achieved. The implementation of this level can be centralized or decentralized, with the second option being the preferred choice when the number of agents is large. Moreover, a distributed implementation in which a node only communicates with a subset of agents (neighbors), defined by the graph, reduces delays in the transmission of signals between the different control levels. As described above, the formation is defined by a set of target distances. That is, if two nodes $v_{i}$ and $v_{j}$ are connected in the graph by an edge, then the distance between them, $d_{ij}$, should reach a desired value $d_{ij}^{*}$. Both the connection between nodes and the target distances $d_{ij}^{*}$ are assumed to be given to ensure the desired dome shape. Additionally, to achieve the semi-spherical surface and guarantee rigidity, all the agents should maintain a distance *R* to the origin.

Let us denote as $p_{i}\in R^{3}$ the position of any agent *i*. In this case, the formation controller is defined as follows: (4)$$u_{i}^{s}=-\sum_{j\in N_{i}}\mu_{ij}({\left(d_{ij}^{*}\right)}^{2}-∥p_{i}-p_{j}{∥}^{2})\left(p_{i}-p_{j}\right)-(R^{2}-∥p_{i}{∥}^{2})p_{i},$$
where $\mu_{ij}>0$ is a gain and $∥p_{i}-p_{j}∥=d_{ij}$ represents the distance between agents *i* and *j*. Note that when all the distances between *i* and its neighbors $j\in N_{i}$ converge to the target values $d_{ij}^{*}$ and the distance of each agent to the origin is *R*, then the control signal $u_{i}^{s}$ approaches zero. For practical reasons, the gains $\mu_{ij}$ are all set to the same value μ<1, which ensures that the second term in (Equation 4) has a higher weight.

From the implementation point of view, the control law (Equation 4) cannot be computed in continuous time but at discrete instances of time. Additionally, each agent needs to obtain the relative distance to its neighbors. Thus, deciding the updating times of (Equation 4) has an impact on both the control performance and the amount of information exchanged through the network. Moreover, the higher the number of nodes *N*, the larger the demand for network usage. For this reason, an event-based communication protocol has been implemented. Event-based sampling is an alternative to the periodic sampling method that has been proven to be effective in reducing the number of samples [44]. The main idea is that it is the state/the output of the system and not the time that determines when to sample the system. In recent decades it has experienced a boom due to the advantages it provides in cyber-physical systems (networked control systems), due to their being resource-constrained.

In this case, the trigger function is defined depending on an error function that depends on the position of the agent and a threshold, so that an event (sampling) occurs when that function reaches that limit value. Usually, the error function e(t) is defined as the norm of the difference between the current measurement of the state x(t) and the last transmitted measurement, $x(t_{k})$. There are different proposals for the threshold in the literature (constant, state-dependent, etc.), but, in general terms, the larger the threshold, the lower the rate of events. Thus, the trigger time of an event-based control system is defined by the following recursive form: (5)$$t_{k+1}=inf\left\{t:t>t_{k},f\left(e\left(t\right),x\left(t\right)\right)>0\right\}$$
where $e\left(t\right)=x\left(t_{k}\right)-x\left(t\right)$ is the error function and f(e(t),x(t)) is the trigger function. The evaluation of the trigger function runs in a loop that, in the current implementation, has a frequency of 50 Hz.

### 3.2. Experiments Description

Once the control objective has been defined, the experiments designed to study the scalability of the tools developed are described below. The different experiments are characterized by an increasing number of agents located along the surface of a hemisphere. A sweep is made from the simplest case with 5 agents to the limits that the simulation tools are able to support, 40 agents. In the hemisphere, the agents are placed at different levels. The level of height z=0 consists of the Khepera IV ground robots moving in the XY-plane. The rest of the agents are of the Crazyflie type and can move in three-dimensional space. Next, we briefly describe the developed experiments:The MRS of experiment A (see Figure 6a,b) consists of a total of five agents, four of which are Khepera IV and one Crazyflie. In this case, all the robots are real and only their corresponding DTs are running in the virtual environment.In experiment B (see Figure 6c,d), the MRS is composed of 10 agents: four Crazyflies, and six Khepera. In this case, four real Crazyflies and four real Kheperas are used. In the virtual environment, two Kheperas run in addition to the virtual twins of the real robots.In experiment C (see Figure 6e,f), the MRS is composed of 15 agents: seven Crazyflies, and eight Khepera. In this case, five real Crazyflies and four real Kheperas are used. The rest of the agents up to 15 are completely digital.The fourth experiment, D (see Figure 6g,h), employs a total of 20 agents, 11 of which are Kheperas and 9 are Crazyflies. In this experience, six Crazyflies and four Kheperas are real. The rest of the agents up to 20 are completely digital.The MRS in experiment E (see Figure 6i,j) is composed by 30 agents. In this case, the distribution of agents is 18 Crazyflies and 12 Khepera.For the last experiment, F, depicted in Figure 6k,l, the number of robots is 40 (26 Crazyflies and 14 Khepera).

From experiences A to C the proportion of real robots is more than 50% (reached in D). From D to F, the number of real robots is maintained at six Crazyflies and four Kheperas, increasing in this way the proportion of digital agents in these experiences progressively. This information is summarized in Table 1 and the final spatial distribution is shown in Figure 6. The video showing the real and virtual environment for the experiment E is available online: https://youtu.be/4H3YZ-sr2mw (accessed on 30 June 2023).

To quantitatively evaluate the experiments, several parameters have been considered. On the one hand, we try to measure how computationally demanding these experiments are when the number of agents increases, and on the other hand, the impact on the performance of the system. These are the performance indices analyzed:Global CPU percentage. This value represents the current system-wide CPU utilization as a percentage.CPU percentage. This represents the individual process CPU utilization as a percentage. It can be >100.0 in case of a process running multiple threads on different CPUs.Real-Time Factor (RTF). This shows a ratio of calculation time within a simulation (simulation time) to execution time (real time).Integral Absolute Error (IAE). This index weights all errors equally over time. It gives global information about the agents.Integral of Time-weighted Absolute Error (ITAE). In systems that use step inputs, the initial error is always high. Consequently, to make a fair comparison between systems, errors maintained over time should have a greater weight than the initial errors. In this way, ITAE emphasizes reducing the error during the initial transient response and penalizes larger errors for longer.

## 4. Results

CPU usage, RTF, and formation error (IAE and ITAE) are used as criteria to compare and assess the simulation performance of Gazebo and Webots for all experiments described in Table 1. In each experiment, the number of agents increases and goes from 5 (experiment A) to 40 (experiment F).

### 4.1. CPU Consumption

Results of the CPU consumption are presented in Figure 7. Figure 7a shows the global CPU usage of the system. Note that this parameter increases with the number of robots for both simulators. However, Gazebo CPU consumption is higher than Webots in all experiments and reaches 100% for N=40 (Experiment F). Indeed, for this later case, Gazebo was unable to run correctly as the timeout of the tool is exceeded when loading the robots at the beginning of the simulation. None of the simulators were able to carry out an experiment with 50 agents. Furthermore, from a performance perspective, a significant difference is observed between the two tools in terms of resource consumption. Figure 7b shows the specific CPU usage for the processes running in each simulation tool, since Gazebo is split into two processes: Gzclient, responsible for running the GUI, and Gzserver, responsible for the physics engine and sensors. The results show that, in all cases recorded, the resource consumption of Gazebo (Gzclient+Gzserver) is more than double that of Webots. It is a clear indicator that Webots is a more efficient tool in terms of CPU usage management. Note that for experiments C, D, and E the threshold of 100% is exceeded for Gazebo’s processes. This just indicates that these processes have more than one thread running.

### 4.2. Real-Time Factor

The Real-Time Factor, RTF, is the ratio between the real execution time and the simulation time. This factor is easily accessible in the simulators. If it reaches the unit value, the simulation is running in real time, and when RTF > 1 it means that the process is running at an accelerated rate. Since we are performing experiments with real and virtual robots, even if all nodes are running in simulated time, this index should be as close to 1 as possible.

Table 2 shows the RTF obtained for each experience. Results are similar for both simulators when the number of agents is under 15. Gazebo achieves good performance in experiment C with 15 agents, but when increasing the number up to 20, the RTF drops below 0.75. In this case, in experiment D (N=20), Webots still maintains a high performance. Experiment E increases the number of agents up to 30, and Webots performance is slightly lower but maintains a value above 0.8, while Gazebo falls below 0.5. Finally, in experiment F, Webots’ RTF drops to a value of 0.56. There is a clear decrease in RTF when the number of robots increases. As Gazebo runs in two processes, when the physics simulation in the successive experiments struggles, Gazebo decreases the RTF sooner, scaling worse than Webots to a large number of robots. Maintaining a high RTF allows running models in real time while using a hardware-in-the-loop simulation to test controllers. In this sense, for experiences as presented in this paper, where batteries in real agents are a limiting factor, the highest RTF is the best. In fact, batteries for Crazyflies can only support 3–5 min flight time, so allowing RTF under 0.75 is not possible.

### 4.3. System Performance

Once metrics related to simulator computational efficiency have been examined, an analysis of convergence times to achieve the desired formation for both simulators is carried out. For this goal, first, the time evolution of the total error weighted by the number of agents for the different experiences is depicted in Figure 8. In a qualitative way, the results are consistent for both simulators, as similar behaviors are found among different tests in all cases. To obtain a quantitative analysis, we compute the IAE and ITAE weighted by the total number of agents and the experiment duration time. The results are shown in Table 3.

Data show that both tools scale well with the number of agents up to the limit supported by each simulator. Indeed, most of the values in Table 3 are in the same range. However, many factors might influence this result, such as the number of digital and real agents, the number of connectivity links, the initial conditions, etc.

## 5. Conclusions

Robots’ simulators are a fundamental part of the development of robotic systems. Therefore, the evaluation and comparison among different simulation systems available is an important task. In this paper, the scalability of Gazebo and Webots simulators integrated with Robotic Park, a MRS-ROS2 experimental platform, has been analyzed. To this end, several experiments are conducted to achieve a rigid formation when the number of agents changes. An event-based communication protocol has been implemented to reduce the demand for network usage, an important issue when the number of robots increases. The combining of physical and virtual robots in both simulators along with ROS 2 has shown an easy and realistic way of increasing the number of agents in the experiments. Models and environments are identical in both tools, facilitating the comparison. The use of resources such as the CPU, the RTF, and the formation error have been evaluated for both tools.

The analyzed simulators present some limitations. For instance, Gazebo does not scale well to a large number of agents, as it presents a limit to the number of robots used in the system by a too-high CPU usage requirement. The results show that Webots obtains the best scores as fewer CPU consumption resources are required when performing the simulation task. It must be noted that there are two processes, client and server, involved in Gazebo due to the implementation design, and without having the agent models integrated. Moreover, RTF, which is considered a good measure of how efficient a simulator is regardless of the hardware it is being run on, has been analyzed. Although the RTF is similar for both tools for a reduced number of agents, it drops when the number of robots increases. This decrease is particularly evident for Gazebo, which is not able to maintain a good RTF above 20 agents.

The results show that Webots allows the combination of a greater size of virtual robots with real robots in collaborative experiences in MRS. Moreover, the consumption of CPU resources for the same number of robots is lower, which highlights its suitability for resource-constrained systems. These benefits, in addition to a friendly user interface for inexperienced users, give Webots added value. Although this study provides some insights into scalability issues of both simulators, further improvements are needed to detail the influence of different factors, such as the type of robots or the control architecture (frequencies, types, onboard versus offboard, etc). Future works will include performance and scalability comparisons of other ROS2-supported simulators or the consideration of other metrics, such as time using different physics engines or the study of the delay impact over the computation. In addition, tests of the simulators under different scenarios involving more complex tasks that require the use of the path planning level will be considered, e.g., in cooperative navigation tasks or with the presence of disturbances.

## Figures and Tables

**Figure 1 sensors-23-06073-f001:**
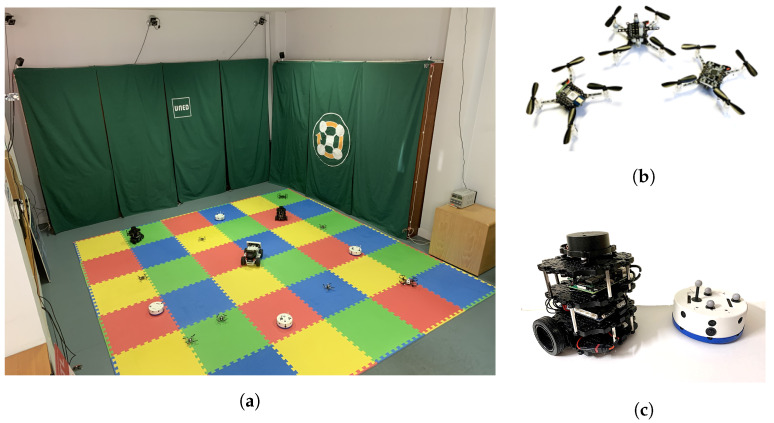
(**a**) Real platform Robotic Park. (**b**) Crazyflies with Loco Positioning deck (**left**), Motion capture marker deck (**center**) and Lighthouse positioning deck (**right**). (**c**) Turtlebot3 Burger (**left**) and Khepera IV (**right**).

**Figure 2 sensors-23-06073-f002:**
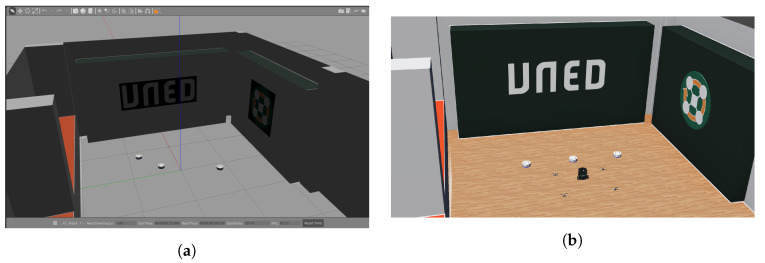
(**a**) Robotic Park in Gazebo. (**b**) Robotic Park in Webots.

**Figure 3 sensors-23-06073-f003:**
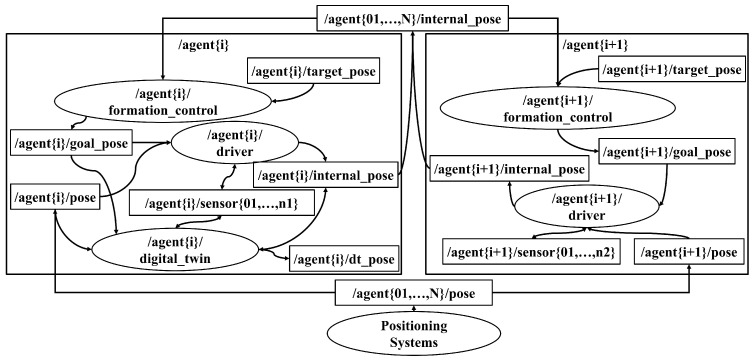
Subset of multi-robot namespace examples. Agent{i}: physical robot with DT; Agent{i + 1}: virtual robot.

**Figure 4 sensors-23-06073-f004:**
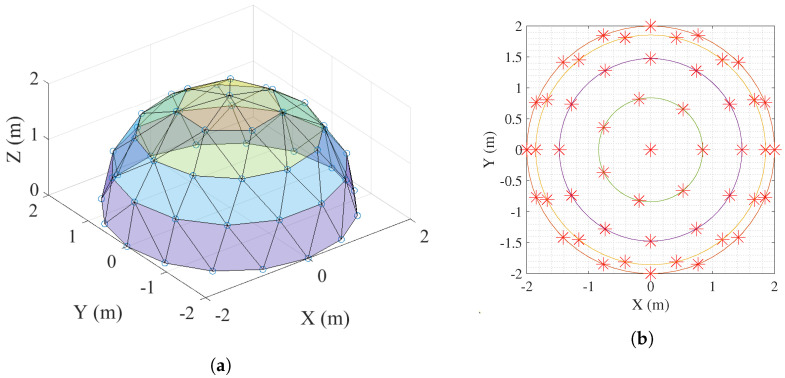
Example of formation with N=50 agents and R=2 m. (**a**) Desired 3D formation. (**b**) Projection over the XY—plane.

**Figure 5 sensors-23-06073-f005:**
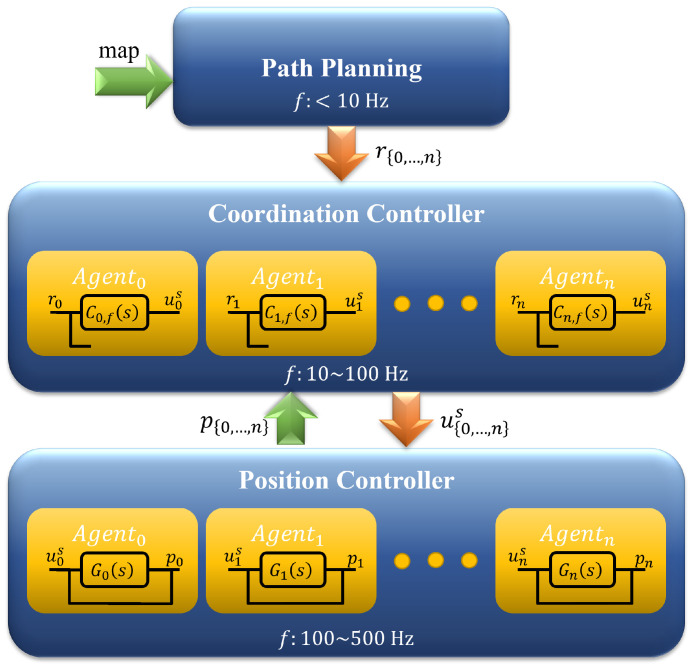
Multi-robot hierarchical control.

**Figure 6 sensors-23-06073-f006:**
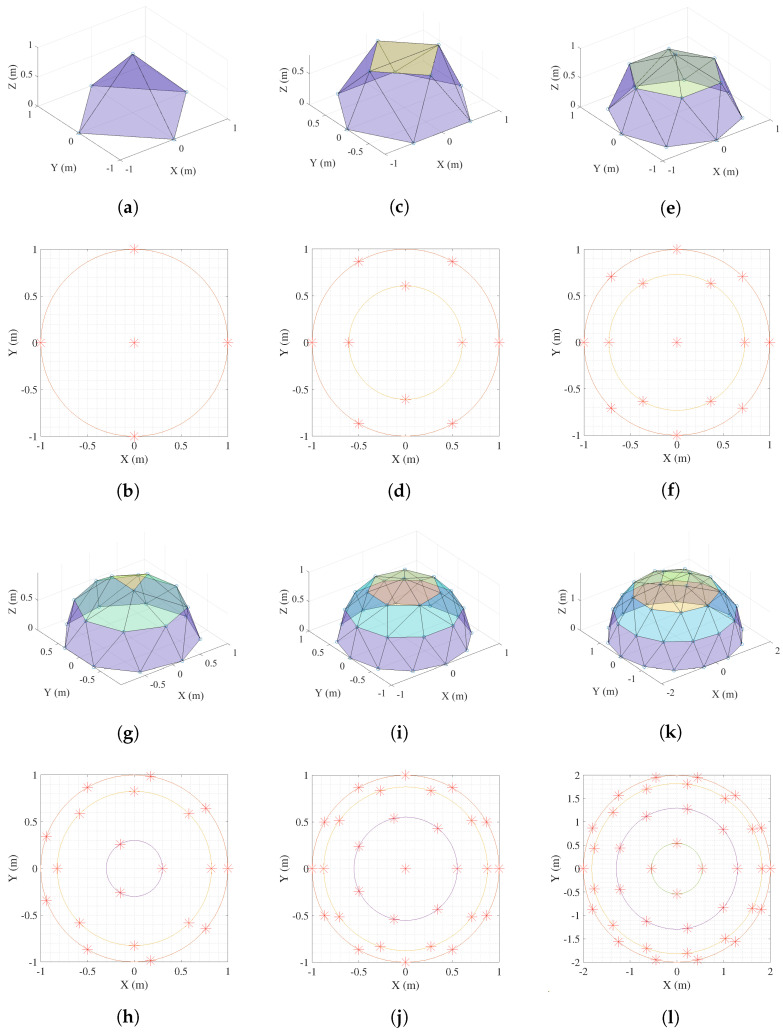
Multi-Robot System formation. Drones are distributed in rings of different height, each of which is drawn with a different color. Experiment A: (**a**) 3D representation, (**b**) 2D agents distribution; Experiment B: (**c**) 3D representation, (**d**) 2D agents distribution; Experiment C: (**e**) 3D representation, (**f**) 2D agents distribution; Experiment D: (**g**) 3D representation, (**h**) 2D agents distribution; Experiment E: (**i**) 3D representation, (**j**) 2D agents distribution; Experiment F: (**k**) 3D representation, (**l**) 2D agents distribution.

**Figure 7 sensors-23-06073-f007:**
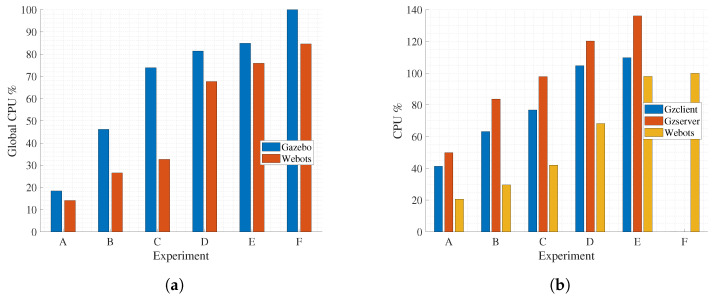
(**a**) General CPU percent usage. (**b**) Simulation tools CPU percent usage.

**Figure 8 sensors-23-06073-f008:**
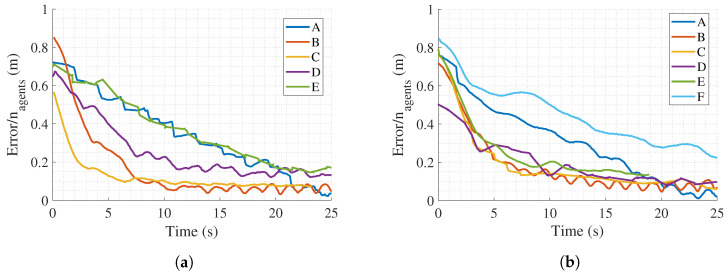
Instant total errors weighted by the total number of agents: (**a**) Gazebo. (**b**) Webots.

**Table 1 sensors-23-06073-t001:** Number of agents for each experiment. The digital twins of real robots are included in the virtual robots column.

			Real Robots		Virtual Robots	
Experiment	Figure	Size	Crazyflie 2.1	Khepera IV	Crazyflie 2.1	Khepera IV
A	Figure 6a,b	5	1	4	1	4
B	Figure 6c,d	10	4	4	4	6
C	Figure 6e,f	15	5	4	7	8
D	Figure 6g,h	20	6	4	11	9
E	Figure 6i,j	30	6	4	18	12
F	Figure 6k,l	40	6	4	26	14

**Table 2 sensors-23-06073-t002:** Real-Time Factor results in Gazebo and Webots.

Experiment	Size	Gazebo	Webots
A	5 agents	0.995	0.977
B	10 agents	0.967	0.977
C	15 agents	0.866	0.962
D	20 agents	0.716	0.941
E	30 agents	0.477	0.831
F	40 agents	-	0.563

**Table 3 sensors-23-06073-t003:** IAE and ITAE in Gazebo and Webots experiences weighted by the total number of agents and experiments duration time.

	IAE (m/s)		ITAE (m)	
Experiment	Gazebo	Webots	Gazebo	Webots
A	0.4485	0.3879	6.8471	3.5558
B	0.3421	0.2481	4.2980	2.5427
C	0.4293	0.3062	5.0444	1.9985
D	0.4619	0.3182	5.0498	2.4613
E	0.5404	0.3108	5.7722	3.5415
F	-	0.6511	-	8.3456

## Data Availability

Data are contained within the article.

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
