# Peer review of "Scalability of Cyber-Physical Systems with Real and Virtual Robots in ROS 2"

_sensors, 2023, doi:10.3390/s23136073_

Round 1

Reviewer 1 Report

The authors evaluate the performance of a cyber-physical system when the number of agents being simulated/controlled increases. The idea is interesting and the paper is well written, with a few minor errors found and listed in sequence, but the authors must address some important points. Plase note that the next comments are intended to improve paper quality and readers' understanding.

At first, I believe the authors must rethink or modify their use of the following terms: VR, AR and MR. The three terms are defined in the text, but in fact none of them are used in this work. At most, the authors are using a simulated virtual environment to experiment with the virtual robots in communication with the real ones, but the definition of MR given does not cover this use at all.

I believe a more appropriate title for this paper would be "Scalability of cyber-physical systems in simulated experiences in ROS 2"

I would not mention AR or MR in this paper, as this work is not directly related to it. Digital twins is a more appropriate term.

I would also remove the term "Mixed reality" from the keyword list as it is not appropriate to this work.

Please provide more detail regarding the experiments performed. The goal activity for all simulated scenarios was robot placement on the surface of a virtual sphere? Please be more clear on that. It would be nice to have more images or even a video showing how the simulations occured.

What is the main focus of the paper? It seems that the authors are comparing two robot simulators (gazebo and webots) but according to Figure 6, for instance, the result is almost obvious: if the number of agents increase, there is a proportional increase in the CPU usage for both simulators.

How is the position of the real robots obtained and passed to the simulation? This is not clear in the text.

Please make clear what are the contributions and limitations of the proposed work. This must be clear in the entire text.

In summary, the authors must provide more information regarding the experiments performed, and clearly state what are the contributions and limitations of the work. What are the benefits to the reader after reading the paper? What important insights can be captured from the paper?

More general comments and minor errors are listed as follows.

"influence to the behavior" -> "influence the behavior"

", Figure 1a." -> "(Figure 1a)."

", Figure 1b," -> "(Figure 1b)"

"IV Figure 1c are" -> "IV (Figure 1c) are"

"flat indoors surfaces." -> "flat indoor surfaces."

"As physic engine," -> "As physics engine,"

" This fact encourage" -> " This fact encourages"

"Digital twin model" -> "A digital twin model"

"D, (see figures 5g and 5h)" -> "D (see figures 5g and 5h),"

"Mixed, Reality (MR) experiments," -> "MR experiments,"

Author Response

Dear Reviewer,

Please find attached the response to all your comments and concerns. Thank you very much for your time and effort in the review.

Best regards,

The authors

Reviewer 2 Report

The Paper evaluates two simulation frameworks used to emulate robot behavior in an environment preconfigured for the experiments.

The authors did an extensive presentation about state-of-the-art and theoretical concepts behind the formation movement of robots in several applicative scenarios: augmented reality, virtual reality, and mixed reality.

The work doesn't match too much with the title, so I propose to change to one more reflecting the paper discussion.

In general, the work is ok, but it needs significant improvement in the presentation of the experiment: it needs to be clarified: how the real robots interact with the simulated, and moreover, which is the goal of the experiment. 

Following some concerns collected during the paper analysis:

  • How are the "path planning" and "Coordination controller" components considered in the loop? 
  • In section 2.4, the authors talk about the initial setup. Why? Is there an evolution of the setup?
  • The authors discuss a mixed usage of digital twins of real devices and simulated robots in the experiments. How does this mixed usage impact in terms of latency in the computation?  

Author Response

(The authors gave the same response as above.)

Round 2

Reviewer 1 Report

Dear authors, thank you for taking into consideration the previous suggestions. I'm satisfied with the modifications performed and believe the paper is ready for acceptance, congratulations!

Please, when preparing the final version of the paper, correct a small text superposition problem in table 1.

Reviewer 2 Report

The authors have addressed most of my previous comments, and about the remaining doubts, the authors answer my concerns.

The title is not satisfying me 100%, but it's more clear what the authors did in this research.

According to question and answer 3: 

In the first round, my hope was that there would be a "not/not clearly" reported description of the "path planning" and "Coordination controller" components, but considering the answer, it is clear that they are not in the loop. This decreases the overall Significance of the Content and the overall merit.